# Platelet-Rich Plasma Gel Matrix (PRP-GM): Description of a New Technique

**DOI:** 10.3390/bioengineering9120817

**Published:** 2022-12-19

**Authors:** Thales Thiago Ferreira Godoi, Bruno Lima Rodrigues, Stephany Cares Huber, Maria Helena Andrade Santana, Lucas Furtado da Fonseca, Gabriel Silva Santos, Gabriel Ohana Marques Azzini, Tomas Mosaner, Chris Paulus-Romero, José Fábio Santos Duarte Lana

**Affiliations:** 1Department of Orthopedics, The Federal University of São Paulo, São Paulo 04024-002, Brazil; 2Brazilian Institute of Regenerative Medicine, Indaiatuba, São Paulo 13334-170, Brazil; 3Brazilian Association for Research in Regenerative Medicine, Indaiatuba, São Paulo 15015-040, Brazil; 4Prime Regen, São Paulo 01308-050, Brazil; 5American Academy and Board of Regenerative Medicine, Lakewood, CO 80401, USA

**Keywords:** platelet-rich plasma, hyaluronic acid, autologous biomaterials, regenerative medicine, orthopedics

## Abstract

Several musculoskeletal conditions are triggered by inflammatory processes that occur along with imbalances between anabolic and catabolic events. Platelet-rich plasma (PRP) is an autologous product derived from peripheral blood with inherent immunomodulatory and anabolic properties. The clinical efficacy of PRP has been evaluated in several musculoskeletal conditions, including osteoarthritis, tendinopathy, and osteonecrosis. When used in combination with hyaluronic acid (HA), a common treatment alternative, the regenerative properties of PRP are significantly enhanced and may provide additional benefits in terms of clinical outcomes. Recently, a new PRP-derived product has been reported in the literature and is being referred to as “plasma gel”. Plasma gels are obtained by polymerizing plasmatic proteins, which form solid thermal aggregates cross-linked with fibrin networks. Plasma gels are considered to be a rich source of growth factors and provide chemotactic, migratory, and proliferative properties. Additionally, clot formation and the associated fibrinolytic reactions play an additional role in tissue repair. There are only a few scientific articles focusing on plasma gels. Historically, they have been utilized in the fields of aesthetics and dentistry. Given that the combination of three products (PRP, HA, and plasma gel) could enhance tissue repair and wound healing, in this technical note, we propose a novel regenerative approach, named “PRP–HA cellular gel matrix” (PRP-GM), in which leukocyte-rich PRP (LR-PRP) is mixed with a plasma gel (obtained by heating the plasma up) and HA in one syringe using a three-way stopcock. The final product contains a fibrin–albumin network entangled with HA’s polymers, in which the cells and biomolecules derived from PRP are attached and released gradually as fibrinolytic reactions and hyaluronic acid degradation occur. The presence of leukocytes, especially monocytes and macrophages, promotes tissue regeneration, as type 2 macrophages (M2) possess an anti-inflammatory feature. In addition, HA promotes the viscosuplementation of the joint and induces an anti-inflammatory response, resulting in pain relief. This unique combination of biological molecules may contribute to the optimization of regenerative protocols suitable for the treatment of degenerative musculoskeletal diseases.

## 1. Introduction

Platelet-rich plasma (PRP) is an autologous product derived from peripheral blood, in which the platelet number is concentrated above the whole blood levels [1]. In the field of regenerative medicine, the biological function of platelets extends beyond hemostasis; their dense granules contain ADP, ATP, serotonin, and calcium, whereas their alpha granules are rich in chemotactic factors, growth factors, and immunomodulatory cytokines [2]. Therefore, upon activation and subsequent degranulation, platelets within the PRP stimulate regenerative processes by promoting vascular remodeling and immunomodulation through the release of signaling molecules [2].

Musculoskeletal disorders are often triggered by an inflammatory process that occurs simultaneously with an imbalance between anabolic and catabolic events [3]. In this regard, PRP may counteract the deleterious effects of excessive tissue inflammation by controlling the same and establishing a pro-anabolic local environment. Consequently, PRP contributes to the restoration of the anabolic–catabolic balance, as it promotes cellular turnover and regulates tissue regeneration via the release of several growth factors, such as the transforming growth factor-β1 (TGF-β1), platelet-derived growth factor (PDGF), vascular endothelial growth factor (VEGF), epidermal growth factor (EGF), etc. [2].

At a transcriptional level, PRP downregulates interleukin-1β (IL-1β), one of the main catabolic cytokines responsible for cartilage degradation [4]. In addition, PRP exerts mitogenic, chemotactic, and proliferative properties; an in vitro study reported that PRP influences chondrocyte activity by promoting anabolic events and increasing the expression of type-II collagen and aggrecans [5]. The clinical efficacy of PRP has been evaluated and established in numerous musculoskeletal conditions, including osteoarthritis, tendinopathies, and osteonecrosis [6,7,8]. PRP is also able to increase extracellular matrix (ECM) synthesis, stimulating wound healing in patients with chronic ulcers [9,10].

Currently, most orthopedic conditions are treated with hyaluronic acid (HA), especially osteoarthritis [11]. HA presents important properties, such as chondroprotection, synthesis of proteoglycans and glycosaminoglycans, and anti-inflammatory and analgesic roles [12]. Chondroprotection is related to the CD44 receptors. A receptor stimulation by HA leads to a decrease in chondrocyte apoptosis (decreasing prostaglandins) in matrix metalloproteinases (MMPs) and IL-1β, as well as to an increase in cell proliferation [13]. Intra-articular injections of HA result in the restoration of the rheological properties of synovial fluid and, when used in combination with PRP, may potentially increase its biological activity, adding to the potential for effective clinical outcomes [11,14,15,16]. The combination of HA and PRP is referred to in the literature as “cellular matrix PRP–HA”. The hypothesis discussed is that the mixture could create a bioactive scaffold around cells that would increase the release time of growth factors [16].

Recently, a new PRP-derived product with viscoelastic properties has been proposed in the literature, named “plasma gel” [17,18]. It is obtained by heating the plasma up to 70 °C for 15–20 min. This process polymerizes the plasmatic proteins, forming a solid thermal aggregate cross-linked with fibrin networks [18]. Plasma gels have been reported to be a rich source of growth factors and to present chemotactic, migratory, and proliferative activities [17,18,19]. Although the number of clinical reports using plasma gels has been increasing over the last few years, its use has been focused on applications in the fields of aesthetics and dentistry [19,20].

Given the broad range of the PRP–HA matrix effects and the function of plasma gels as natural scaffolds able to deliver bioactive molecules, here, we describe a novel technique that combines leukocyte-rich PRP, HA, and autologous plasma gels as a valuable biological tool in the application of regenerative medicine techniques.

## 2. Technique and Processing Method

### 2.1. Obtaining Leukocyte-Rich PRP

To obtain leukocyte-rich PRP (LR-PRP), 60 cc of peripheral blood was collected from a patient under aseptic conditions. To better preserve the blood cells and maintain a physiological pH [21], the blood was drawn into 8.5 mL tubes containing the anticoagulant ACD (citric acid, sodium citrate, and dextrose) (Becton, Dickinson, and Company (BD), Franklin Lakes, NJ, USA).

The variables of time and force of centrifugation in cell recovery are relevant parameters that should be taken into account in order to obtain a high quality PRP product, which has been a subject of study by our group [22,23,24]. In that manner, the protocol proposed consists of two centrifugation steps, performed in falcon tubes. The first one, at 300× *g* for 5 min, separates the blood components into three distinct layers in order of increasing density as follows: plasma is located at the uppermost layer, followed by the buffy coat, which is a platelet- and leukocyte-rich thin layer, and erythrocytes are at the bottom (Figure 1A). The plasma and the buffy coat were collected into another tube for the second centrifugation, at 700 × *g* for 17 min (Figure 1B). At this higher centrifugation force, the platelets and the leukocytes form a proper sediment at the bottom of the tube, thus, leading to a more concentrated product (Figure 1D). Eighty percent of the upper plasma layer represents the platelet-poor plasma (PPP), which is collected to obtain the plasma gel. Twenty percent of the remaining volume represents the leukocyte-rich PRP (Figure 1E), which is collected and homogenized to be eventually mixed with the plasma gel (Figure 1F).

### 2.2. Obtaining the Plasma Gel

Plasma gels were first reported by Everts et al. [17] in 2016 and obtained by heating up the upper layer of the PRP to 70 °C for approximately 15 min. The increase in the temperature polymerizes the albumin, which represents the most abundant plasma protein, and forms aggregates that are cross-linked with fibrin networks (Figure 2) [18].

The PRP effects, such as immunomodulation and wound repair, could be enhanced when injected with a plasma gel, as its 3-dimensional structure (resulting from the heating) allows for the migration and proliferation of the platelets and leukocytes present in the PRP. In addition, the fibrinolytic reactions potentiate such effects and also assist in the homing and activation of mesenchymal stem cells [2].

To obtain a proper matrix, an efficient dispersion of the components must be provided to avoid phase separation. This dispersion can be prepared using a three-way stopcock device, in which a syringe containing the leukocyte-rich PRP is attached to one side and a syringe containing the plasma gel is attached to the other side, as well as HA, which is also added on the third valve (Figure 3E). The device allows for a proper mixture and keeps the components in a single syringe, ready to be injected (Figure 3F).

### 2.3. The PRP Gel Matrix (PRP-GM)

The dispersion of the plasma gel in the PRP–HA matrix forms a semi-network of the polymerized fibrin fibers, which contain the plasma gel inside their pores. The resulting biomaterial matrix with less porosity should offer a greater barrier, promoting the slower release of growth factors compared to PRP [16,18]. In addition, the dispersion of the plasma gel should delay gelation, and the topography of the PRP–HA matrix should have a positive influence on cell adhesion and macrophage polarization. Additionally, the denser network packing provided by the plasma gel should improve the rheological properties. While the scientific literature has documented the properties of HA as a potent biological product for the treatment of osteoarthritis [25], the cellular gel matrix PRP–HA could provide better performance than leukocyte-rich PRP (LR-PRP) or HA alone (Figure 4). Figure 5 represents a schematic method for obtaining PRP-GM, and Table 1 summarizes the steps of the procedure.

## 3. Discussion

PRP was first used to treat thrombocytopenia [26]. However, due to its regenerative properties, its use has been extended to multiple medical and surgical fields, including skin rejuvenation and hair regrowth [27,28], cardiac surgery [29], lung dysfunction [30], neural repair [31], and orthopedics [14]. For the latter, several meta-analysis studies have reported PRP as a safe and effective treatment to address tendon injuries [32], osteoarthritis [33], and meniscal lesions [34].

The presence of leukocytes in PRP is a matter of discussion in the literature [35]. With regard to mononuclear cells, macrophage plasticity is a feature that influences tissue repair. The ability to switch phenotypes, either M1 or M2 (depending on local molecular signals as a source of stimuli), has provided macrophages with the ability to compete with platelets as the real workers in PRP therapies [35]. The transition between both subtypes is mediated by the late stages of wound healing. At this stage, the M1 subtype is stimulated by microbial agents and can be considered a pro-inflammatory cell, promoting the apoptosis of neutrophils and the clearance of these cells. The M2 subtype is associated with tissue repair as it presents anti-inflammatory properties by secreting the bone morphogenetic proteins TGF-β, PDGF, and insulin-like growth factor (IGF) in areas of inflammation or injury in different tissues in the body. This promotes the necessary recruitment and proliferation of osteoblasts, stem cells, and progenitor cells [2,35].

The hemostatic functions of platelets are also relevant factors that may enhance the PRP wound repair effect. Once the platelets are activated, there is a release of coagulation factors that promote the polymerization of fibrin monomers and, consequently, the formation of a clot. In addition to preventing blood loss, the clot acts as a provisional matrix as the cross-linked fibrin fibers allow for cell migration during the repair process [36]. The clot also works as a reservoir of signaling molecules that are released upon clot degranulation [37].

Fibrinolysis also exerts a role in tissue repair, as it promotes cell migration by the release of growth factors and regulates protease systems involved in regeneration, such as the urokinase plasminogen activator receptor (uPAR) and the plasminogen activator inhibitor-1 (PAI-1); both systems are vital for the recruitment and activation of mesenchymal stem cells [2,38]. In this manner, several studies have proposed the addition of calcium chloride to PRP in order to promote platelet activation, resulting in an autologous clot. This approach is commonly used for the treatment of ulcers [39,40,41].

The use of HA as a scaffold presents important properties, such as biodegradability, biocompatibility, and bioresorbability [42]. De Melo et al. (2020) described a semi-interpenetrating polymer network (semi-IPN), formed by fibrin–HA. The formation is defined as a network composed of polymers in both cross-linked and linear forms. This specific structure is formed by polymerized fibrin and HA coils entangled among the fibers. HA binds to specific fibrin sites through physical cross-linking but without covalent bonds, with heterogeneous packing arrangements and viscoelasticity, which varies according to HA’s molecular mass and concentration. As a consequence, a fibrin polymerization delay occurs in the mixture of HA–LP-PRP. On the other hand, the presence of HA is able to entrap more leukocytes within the fibrin–HA structures, which could interact with and affect biological processes [16].

The interaction between HA and PRP has a great influence on clinical outcomes. Molecular studies have reported a synergistic anabolic effect on cartilage regeneration, as the combination of both products can induce the expression of chondrogenic markers, such as SOX 9 and COLL II, and decrease the expression of inflammatory markers, including IL1B, MMP1, and MMP3 [43]. Clinical evidence has also supported the combined use of HA and PRP. Clinical trials have reported a higher beneficial effect on pain relief and improvements in joint functions when both products were used together rather than using PRP or HA alone [14,44,45,46].

With regard to the plasma gel, the results from rheological and mechanical evaluations identify it as an ideal biomaterial to be used in combination with PRP, as it is considered to be a non-cytotoxic product suitable for cell growth that presents the ability to release growth factor loads progressively over time [18]. In addition, there is a nutrition component with the addition of proteins and the filling role of the autologous gel biological product.

We, therefore, propose a novel technique that combines PRP, HA, and plasma gels—the PRP–HA cellular gel matrix (PRP-GM). Given the regenerative properties cited above, the PRP-GM could be used as a powerful tool for the treatment of degenerative musculoskeletal diseases.

## 4. Conclusions

The use of PRP combined with HA has been widely discussed in the literature. The use of HA alone is one of the most common approaches to treating musculoskeletal conditions, especially osteoarthritis. However, by itself, it does not promote cartilage regeneration. PRP represents an alternative tool to treat degenerative conditions. Here, we described a new technique to obtain a combination of both products, called the “PRP matrix”. The addition of an autologous plasma gel could increase the regenerative property of the PRP matrix, as clot formations and fibrinolytic reactions exert an important role in tissue regeneration. Given the viscoelasticity characteristics of plasma gels and their interactions with the PRP content, the addition of HA promotes an increase in the viscosupplementation of damaged tissues, besides its anti-inflammatory function. Therefore, the use of the cellular gel matrix containing HA–LR-PRP and plasma gel (PRP-GM) could present an optimization of PRP protocols suitable for use in regenerative applications.

## Figures and Tables

**Figure 1 bioengineering-09-00817-f001:**
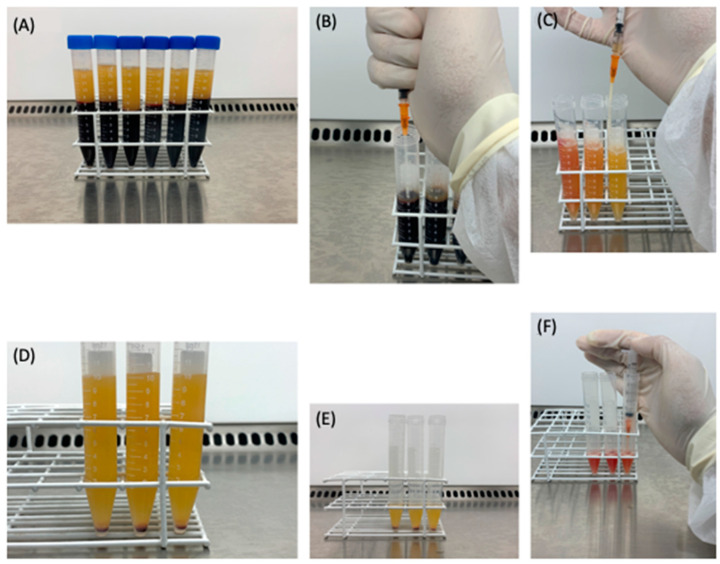
Processing of the LR-PRP. (**A**) Separation of the blood components, resulting from the first centrifugation; (**B**) Collection of the plasma and buffy coat; (**C**) Plasma and buffy coat, added in another falcon tube; (**D**) Buffy coat-containing pellet after the second centrifugation; (**E**) Twenty percent of the bottom layer, representing the LR-PRP; (**F**) Homogenized LR-PRP.

**Figure 2 bioengineering-09-00817-f002:**
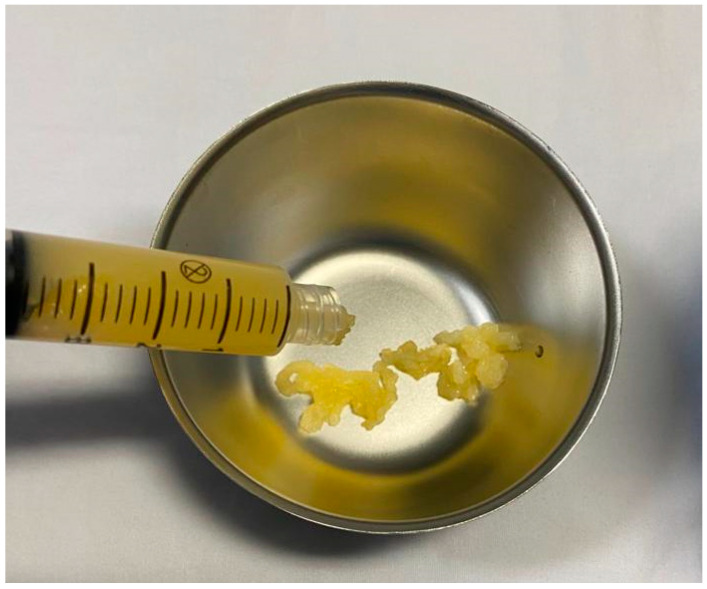
Heat-induced plasma gel formation.

**Figure 3 bioengineering-09-00817-f003:**
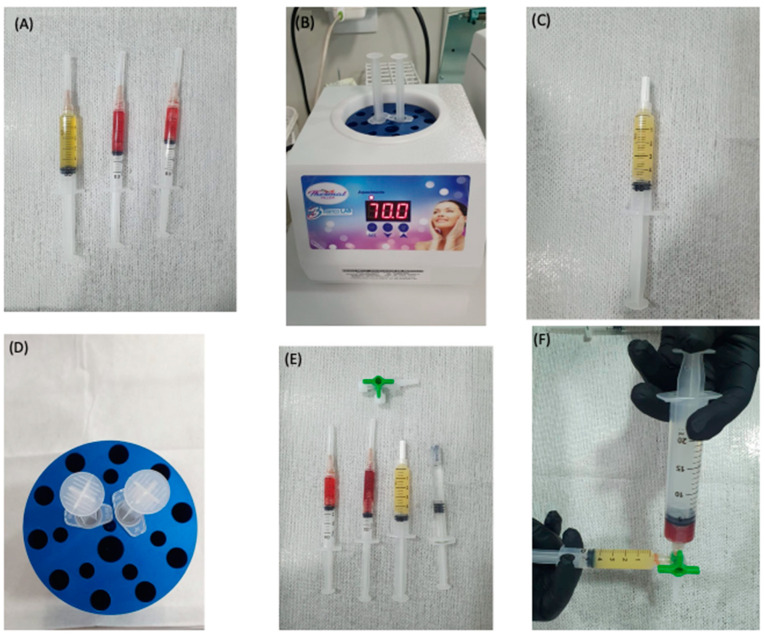
Obtaining the PRP–HA cellular gel matrix. (**A**) Syringes containing the plasma and buffy coat; (**B**) Plasma being heated up to 70 °C for 15 min; (**C**) The plasma gel; (**D**) The cooling device; (**E**) The buffy coat, plasma gel, hyaluronic acid, and three-way connector before mixing; (**F**) Mixing of the buffy coat with the plasma gel.

**Figure 4 bioengineering-09-00817-f004:**
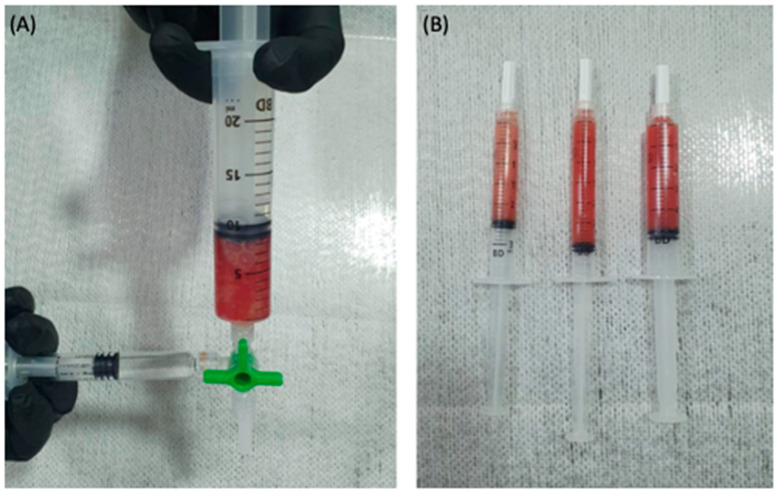
Last step before infiltration. (**A**) The mixed buffy coat and plasma gel are homogenized with hyaluronic acid to obtain the PRP–HA cellular gel matrix; (**B**) The final product, ready for use.

**Figure 5 bioengineering-09-00817-f005:**
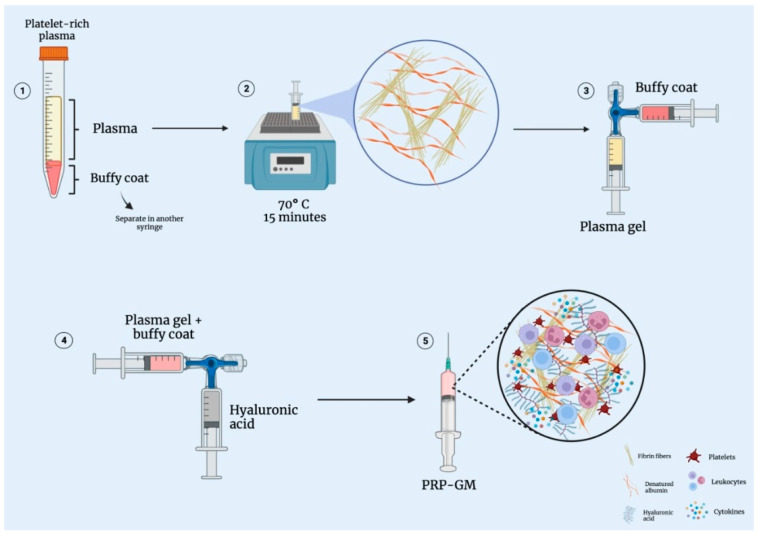
Illustrated chart representing the method for obtaining the PRP-GM. (**1**) PRP is prepared as previously described [10]; (**2**) The collected plasma is heated up to 70 °C for 15 min in order to form a gel; (**3**) The plasma gel and buffy coat are mixed into a syringe; (**4**) Hyaluronic acid is added to the mix; (**5**) Schematic illustration of all of the components present in the PRP-GM.

**Table 1 bioengineering-09-00817-t001:** Methods and materials to obtain the PRP-GM.

	Procedure
LR-PRP	Harvest peripheral blood in ACD tubes
	First spin at 300× *g* for 5 min (RT)
	Collect plasma and buffy coat
	Second spin at 700× *g* for 17 min (RT)
	Collect 80% of the supernatant for plasma gel in a syringe
	Homogenize and separate the remaining 20% (LR-PRP)
Plasma gel	Place the syringes containing the plasma in a heating device
	Heat up to 70 °C for 15 min
Hyaluronic acid	Use commercial hyaluronic acid
PRP-GM	Mix LR-PRP, plasma gel and hyaluronic acid into a single syringe
	Inject into the target point using an ultrasound guidance

LR-PRP, leukocyte-rich platelet-rich plasma; RT, room temperature; PRP-GM, platelet-rich plasma gel matrix.

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
