# Peer review of "Platelet-Rich Plasma Gel Matrix (PRP-GM): Description of a New Technique"

_bioengineering, 2022, doi:10.3390/bioengineering9120817_

Round 1

Reviewer 1 Report

Well written technical report on improving the efficacy of PRP.

I have one comment, Page 6, line 150:

Abbreviation "LR" in LR-PRP stands for what?

Author Response

Reviewer 1

I have one comment, Page 6, line 150. Abbreviation "LR" in LR-PRP stands for what?

Answer: We’d like to thank the reviewer for pointing out the lack of the meaning of LR. The initials stand for leukocyte-rich. We have added the meaning in the line 150 before the initials and also in the line 96 from the page 4, where the word first appeared.

Reviewer 2 Report

The subject matter of this technical note deals with the preparation of “PRP-HA cellular gel-matrix” (PRP-GM) composed of ternary biological molecules, i.e. PRP, HA, and plasma gel.

The manuscript itself is considered to be theoretically and structurally reasonable. The present study is well worth investigating, and the authors did well in stating what the goal of the paper is.

However, if the authors could clear some minor issues as shown below, this paper seems to be qualified to ensure its publication.

It is recommended that the authors provide the followings:

1) a schematic that shows the entire process of preparing PRP-GM at a glance and

2) a summary protocol for each step of the preparation process (including specific conditions and apparatus/device settings; e.g., temperature, volume, concentration, pressure, rpm, etc.).

Author Response

Reviewer 2

The subject matter of this technical note deals with the preparation of “PRP-HA cellular gel-matrix” (PRP-GM) composed of ternary biological molecules, i.e. PRP, HA, and plasma gel.

The manuscript itself is considered to be theoretically and structurally reasonable. The present study is well worth investigating, and the authors did well in stating what the goal of the paper is.

However, if the authors could clear some minor issues as shown below, this paper seems to be qualified to ensure its publication.

It is recommended that the authors provide the followings:

1) a schematic that shows the entire process of preparing PRP-GM at a glance and

2) a summary protocol for each step of the preparation process (including specific conditions and apparatus/device settings; e.g., temperature, volume, concentration, pressure, rpm, etc.).

Answer: We’d like to thank the reviewer for the comments. A schematic was added in the manuscript summarizing all the process to obtain PRP-GM (figure 4). In addition, we have added a table (table 1) listing the steps to get the final product.

Reviewer 3 Report

The authors reported a study where they have proposed a method which help in the optimization of regenerative protocols suitable for the treatment of degenerative musculoskeletal diseases; by using a combination of biological molecules, they have developed a protocol to obtaining these new materials. The study had an interesting idea with high possibility to be applied. However, there are issues to clarify so that the scientific material to be clearly understood. Please find below some comments/suggestions which might help to improve the quality of the manuscript.

1.      Abstract section contains general introductory and from literature issues and less a brief presentation of the study; the authors are requested to reformulate abstract in the way to stress out the outcome of the study, methods to be used, brief results and conclusion.

2.      The authors stated that they bring as novel aspect a “valuable biological tool”; however, details on this tool were not found or mentioned.

3.      Experimental section should be organized in such way to be clear for the readers: materials, preparation methods, characterization methods, etc. The authors are requested to complete the missing information.

4.      Section 2.2. was entitled “Plasma gel”; however, the exact meaning of this section was not fully understood; the authors are advised to reformulate or to insert further clarifications.

5.      The authors mixed some constituents, and no composition was shown together with all the constituents of the final gel. The authors are requested to provide details so that the material to be clearly understood.

6.      The proof of blending crosslinking or another type of bonding or combination has not been proven by a spectrum or any other king of quantification analysis. The authors mention in the introduction about the gels showing viscoelastic properties; the authors are requested to include experimental analysis to prove their outcome except some photographs.

7.      Conclusion part should be reformulated accordingly after proving the missing information.

Author Response

Reviewer 3

The authors reported a study where they have proposed a method which help in the optimization of regenerative protocols suitable for the treatment of degenerative musculoskeletal diseases; by using a combination of biological molecules, they have developed a protocol to obtaining these new materials. The study had an interesting idea with high possibility to be applied. However, there are issues to clarify so that the scientific material to be clearly understood. Please find below some comments/suggestions which might help to improve the quality of the manuscript.

  1. Abstract section contains general introductory and from literature issues and less a brief presentation of the study; the authors are requested to reformulate abstract in the way to stress out the outcome of the study, methods to be used, brief results and conclusion.

Answer: We thank the reviewer for the suggestion. We tried to expand more on the content of the product and how it is obtained as well as its main function as a regenerative tool in the abstract section.

  1. The authors stated that they bring as novel aspect a “valuable biological tool”; however, details on this tool were not found or mentioned.

Answer: We’d like to thank the reviewer of the comment. By “valuable biological tool” we suggest that the LR-PRP, with all its well-known functions, combined with plasma gel can further contribute to tissue regeneration. The roles of the components that make up this orthobiologic product have been well documented in the literature. For instance, PRP, a well-known biomaterial, releases a wide array of active biomolecules, including a variety of growth factor and anti-inflammatory cytokines that enhance tissue repair by promoting vascular remodeling and modulating escalated inflammatory processes (reference 2). Fibrin fibers present in plasma gel are also a rich source of biomolecules with chemotactic, migratory, and proliferative activities (references 17 – 19). Furthermore, fibrinolytic reactions are known to dictate the recruitment and activation of mesenchymal stem cells to the site of injury, allowing tissue repair (references 2 and 38). Hyaluronic acid is another tool which works well with PRP, thus contributing to clinical effects. It represents an ordinary alternative for treating degenerative cartilage and soft tissue diseases via viscosupplementation and anti-inflammatory responses, resulting in pain relief (reference 12).

  1. Experimental section should be organized in such way to be clear for the readers: materials, preparation methods, characterization methods, etc. The authors are requested to complete the missing information.

Answer: Dear reviewer 3, we thank you for your kind observation.  We apologize if the section generated confusion. The procedure for obtaining the gel and how to use it was better detailed with figure 4 and table 1, providing clarification which speaks directly to the materials and methods.

  1. Section 2.2. was entitled “Plasma gel”; however, the exact meaning of this section was not fully understood; the authors are advised to reformulate or to insert further clarifications.

Answer: We’d like to thank the reviewer of the comment. The section was supposed to explain how to obtain plasma gel. We have changed the section title to “Obtaining plasma gel”. In addition, we have added the sentence “Plasma gel was first reported by Everts et al. [17] in 2016 and it is obtained by heating the upper layer of the PRP up to 70°C for approximately 15 minutes. The increase of temperature polymerizes albumin, which represents the most abundant plasma protein, and forms aggregate crosslinked with fibrin networks [18]“ to this section.

  1. The authors mixed some constituents, and no composition was shown together with all the constituents of the final gel. The authors are requested to provide details so that the material to be clearly understood.

Answer: We’d like to thank the reviewer of the comment. The composition of PRP-GM is based on previous investigations from our own group, dear colleagues and other institutions. The composition of LR-PRP has been widely in the literature. Platelets’ granules release its biomolecules, including growth factors, cytokines and ADP, once activated (reference 2). The presence of leukocytes in PRP has been a matter of discussion. However, we support the hypothesis that they can positively influence tissue healing given that the type 2 macrophage exerts anti-inflammatory function (reference 35). The mix of LR-PRP with hyaluronic acid results in an interpenetrating fibrin network with HA’s polymers, leading to the formation of a cell-friendly hydrogel scaffold rich in soluble factors, favoring mesenchymal stem cells with superior chondrogenesis and osteogenesis capabilities compared to PRP alone (reference 16). The upper layer fraction of the plasma is composed by plasmatic proteins. After a heating process, these proteins undergo conformational alterations and give rise to solid thermal aggregates enclosed by a stable fibrin network that allow gradual growth factor release (reference 18).

  1. The proof of blending crosslinking or another type of bonding or combination has not been proven by a spectrum or any other king of quantification analysis. The authors mention in the introduction about the gels showing viscoelastic properties; the authors are requested to include experimental analysis to prove their outcome except some photographs.

Answer: We’d like to thank the reviewer of the suggestion. Considering the structures of the plasma gel-fibrin and HA-fibrin binary systems, we can infer that the ternary structure of the plasma gel matrix of this work at the volumetric proportion 5:2:2 forms structures similar to the binary ones, and HA interacts mainly with the fibrin network formed in the pores of the plasma gel primary structure. Analyzing in terms of viscoelasticity (G' modulus) (references 16, 18,19), plasma gel has the stiffest structure followed by the fibrin network and HA. Therefore, the lower stiffness of the HA and fibrin structures justifies their preferential interaction and the inferred structure. As the technical note of this work aims to present at first hand the feasibility of preparing this new tool, the 3-component matrix prepared in a three-way system, we only infer about the ternary structure formed, based on previous works. Considering the importance of interactions in the new matrix, structural and rheological characterizations are underway in our group.

               An experimental analysis of blending crosslinking was proved in our previous work (reference 16) by Fourier-transform infrared spectroscopy (FTIR) spectra of the semi-interpenetrating fibrin networks from LR-PRP and high molecular weight HA in a ratio of 1:1 v/v. In addition, scanning electron microscopy (SEM) images showed structures and morphologies different for fibrin and the fibrin-HA network. Rheological analysis showed the interpenetrating networks of fibrin-HA were softer, with lower viscoelasticity modulus, compared to the fibrin hydrogel. Fibers from the fibrin hydrogel were thin and elongated with an average diameter of 132 ± 35 nm, while thicker fibers with an average diameter of 225 ± 50 nm were arranged in an inhomogeneous matrix and in more closely packed domains in the LR-PRP-HA hydrogels.  Regarding the interactions between the plasma gel and PRP, SEM images from a study published by Anitua et al. (reference 18) show a porous structure of large patches of protein aggregates, with the fibrin network mainly formed inside the gel matrix pores.

  1. Conclusion part should be reformulated accordingly after proving the missing information.

Answer: We’d like to thank the reviewer of the suggestion. According to the answer provided in the previous question, we have added the sentence “Given the viscoelasticity characteristic of the plasma gel and its interaction with PRP content, the addition of HA promotes an increase of viscosupplementation of the damages tissue, besides its anti-inflammatory function” to the conclusion section.

Thank you very much for taking the time to analyze our manuscript and proposing thoughtful considerations.

Round 2

Reviewer 3 Report

Dear Editor, Authors,

Thank you for the provided answers. They have partly answered my queries. Additionally, from the provided answers, it was understood that the main work presented as it should be was completely described within the authors' previous work so that the reason for publishing a method and results already presented was not clear.  To avoid duplicates, the authors should provide clear justification  and the difference between the present work and work presented within ref 16.  
